# Health services costs for ovarian cancer in Australia: Estimates from the 45 and Up Study

**David E. Goldsbury** [1] *, **Amy Vassallo** [2,3,4] *, **Marianne F. Weber** [1], **Julia Steinberg** [1], **Penelope M. Webb** [5], **Anna DeFazio** [1,6,7,8], **Karen Canfell** [1,7,9]

**1** The Daffodil Centre, The University of Sydney, a Joint Venture with Cancer Council NSW, Sydney, Australia, **2** The George Institute for Global Health, UNSW, Sydney, Australia, **3** School of Public Health, Faculty of Medicine and Health, The University of Sydney, Sydney, Australia, **4** Past Affiliation Cancer Research Division, Cancer Council NSW, Sydney, Australia, **5** Population Health Department, QIMR Berghofer Medical Research Institute, Brisbane, Qld, Australia, **6** Department of Gynaecological Oncology, Westmead Hospital, Sydney, NSW, **7** Faculty of Medicine and Health, University of Sydney, Sydney, NSW, Australia, **8** Centre for Cancer Research, The Westmead Institute for Medical Research, Sydney, NSW, Australia, **9** Prince of Wales Clinical School, UNSW Medicine, Sydney, NSW, Australia

* dgol6982@sydney.edu.au (DEG); avas0473@uni.sydney.edu.au (AV)

## Abstract

### Introduction

There have been significant advancements in risk identification and treatment for ovarian cancer over the last decade. However, their impact on health services costs is unclear. This study estimated the direct health system costs (government perspective) for women diagnosed with ovarian cancer in Australia during 2006–2013, as a benchmark prior to opportunities for precision-medicine approaches to treatment, and for health care planning.

### Methods

Using cancer registry data, we identified 176 incident ovarian cancers (including fallopian tube and primary peritoneal cancer) in the Australian 45 and Up Study cohort. Each case was matched with four cancer-free controls on sex, age, geography, and smoking history. Costs were derived from linked health records on hospitalisations, subsidised prescription medicines and medical services to 2016. Excess costs for cancer cases were estimated for different phases of care relative to cancer diagnosis. Overall costs for prevalent ovarian cancers in Australia in 2013 were estimated based on 5-year prevalence statistics.

### Results

At diagnosis, 10% of women had localised disease, 15% regional spread and 70% distant metastasis (5% unknown). The mean excess cost per ovarian cancer case was $40,556 in the initial treatment phase (≤12 months post-diagnosis), $9,514 per annum in the continuing care phase and $49,208 in the terminal phase (up to 12 months before death). Hospital admissions accounted for the greatest proportion of costs during all phases (66%, 52% and 68% respectively). Excess costs were higher for patients diagnosed with distant metastatic disease, particularly during the continuing care phase ($13,814 versus $4,884 for localised/

**Data Availability Statement:** The data cannot be made available by the authors. These are third party data not owned or collected by the authors and on-provision is not permitted by the relevant data custodians (Sax Institute, Services Australia,

NSW Ministry of Health), as it would compromise the participants' confidentiality and privacy. The data contain potentially identifying and sensitive patient information. However the data are available from the data custodians for approved research projects - data access enquiries can be made to the Sax Institute (see https://www.saxinstitute.org.au/our-work/45-upstudy/governance/ for details). Other researchers would be able to access these data using the same process followed by the authors.

**Funding:** AV received a postdoctoral fellowship funded by the Fussell Family Foundation to support work relating to ovarian cancer. No funder had a role in study design, analysis, decision to publish, or preparation of the manuscript. PMW received grant funding from AstraZeneca for an unrelated study of ovarian cancer, and a speaker's fee from AstraZeneca. ADF received grant funding from AstraZeneca for an unrelated study of ovarian cancer. DEG, MFW, JS and KC are investigators on a study of cancer patient population projections, funded by the Australian Government Medical Research Future Fund (MRFF) – Preventive and Public Health Research Initiative – 2019 Targeted Health System and Community Organization Research Grant Opportunity (MRF1200535). KC receives salary support from the National Health and Medical Research Council Australia (APP1194679). KC is co-PI of an investigator-initiated trial of cervical screening, "Compass", run by the Australian Centre for Prevention of Cervical Cancer (ACPCC), which is a government-funded not-for-profit charity. Compass receives infrastructure support from the Australian government and the ACPCC has received equipment and a funding contribution from Roche Molecular Diagnostics, USA. KC is also co-PI on a major implementation program Elimination of Cervical Cancer in the Western Pacific which has received support from the Minderoo Foundation and the Frazer Family Foundation and equipment donations from Cepheid Inc. Neither KC nor their institution (the Daffodil Centre, a joint venture between Cancer Council NSW and The University of Sydney) receive direct funding from commercial organisations.

**Competing interests:** PMW received grant funding from AstraZeneca for an unrelated study of ovarian cancer, and a speaker's fee from AstraZeneca. ADF received grant funding from AstraZeneca for an unrelated study of ovarian cancer. KC is co-PI of an investigator-initiated trial of cervical screening, "Compass", run by the Australian Centre for Prevention of Cervical Cancer (ACPCC), which is a government-funded not-for-profit charity. ACPCC

regional disease). The estimated overall direct health services cost of ovarian cancer in 2013 was AUD$99million (4,700 women nationally).

## Conclusion

The excess health system costs of ovarian cancer are substantial. Continued investment in ovarian cancer research, particularly prevention, early detection and more effective personalised treatments is necessary to reduce the burden of disease.

## Introduction

While ovarian cancer is a less common cancer in terms of incidence, its social and health system impacts are substantial. Globally there were an estimated 313,959 new cases of ovarian cancer diagnosed in 2020 and 207,252 deaths (1.6% of all cancers, with a higher proportion of cancer deaths, 2.1%) [1]. In Australia, ovarian cancer is the fifth leading cause of cancer death amongst women [2], with an estimated 1,703 new tubo-ovarian cancer diagnoses and 1,037 deaths in 2020 [2].

Due to its non-specific symptoms, propensity for rapid widespread intra-peritoneal metastasis and lack of an effective early detection test [3], ovarian cancer is often detected at an advanced stage, making curative treatment challenging. Primary treatment during the study period typically involved cytoreductive surgery combined with adjuvant or neoadjuvant platinum-based chemotherapy [4], with bevacizumab maintenance treatment added for women at high risk of recurrence from 2014 [5]. Over 80% of ovarian cancer patients will experience disease relapse [6], and eventual resistance to chemotherapy is common [7]. Additional health care interventions for disease surveillance can include regular blood tests, scans, physical examinations or surgery for recurrence [8]. Treatment with poly-ADP ribose polymerase (PARP) inhibitors (PARPi, such as olaparib) has been shown to be effective [9] and in Australia is reimbursed by government for maintenance front-line treatment (since 2020) and in the relapse setting (since 2017) in women with pathogenic variants in the *BRCA1* or *BRCA2* gene [10, 11].

Diagnosis at an advanced stage and ongoing health interventions for surveillance and relapse mean that diagnosis and interventions for ovarian cancer result in significant health care costs [12–15]. For example, a previous analysis of United States claims data, from Commercial Claims and Encounters and Medicare, found that ovarian cancer had the highest per case treatment costs out of 10 different cancers in 2000, with almost US$52,000 worth of attributable costs per case over a mean follow-up of 12 months [12].

In Australia, health care is largely funded through national and state governments, and delivered through Medicare, a universal health insurance scheme. A component of Medicare is the Medicare Benefits Schedule (MBS) which reimburses for activities such as consultations with general practitioners and specialist doctors, imaging, pathology tests, and selected procedures. Another component, the Pharmaceutical Benefits Scheme (PBS), subsidises registered and indicated medications. Health care accounts for a substantial portion of the Australian budget, and our recent study investigating the health system costs of a range of cancers estimated that the cost of cancer to the health system was at least $6.3billion in 2013. This cost is expected to grow as both number of cases and survival increase [16, 17]. Our previous Australian study of women diagnosed with ovarian cancer in 2003–2005, reported a mean 2008AUD $48,000 in costs per case over 2.5 years (n = 85, 74% FIGO stage III at diagnosis); however, this

has received equipment and a funding contribution from Roche Molecular Diagnostics, USA. KC is also co-PI on a major implementation program Elimination of Cervical Cancer in the Western Pacific which has received equipment donations from Cepheid Inc. There are no patents, products in development or marketed products associated with this research to declare. This does not alter our adherence to PLOS ONE policies on sharing data and materials.

study only considered women referred for chemotherapy and health service costs over the first 2.5 years after diagnosis [18].

Benchmarking treatment and associated costs is vital, particularly in periods surrounding substantial changes to clinical practice, as cancer cost data are needed to evaluate any population-level intervention for cancer treatment, detection or prevention. As the ovarian cancer clinical landscape continues to evolve, we sought to benchmark treatment costs in the period just prior to the implementation of some substantial changes in Australia (including the use of bevacizumab and PARPi). These benchmark data can be used for future evaluations of these new treatment interventions and/or future approaches to prevention and detection, as well as help understand the implications of disruptions to health services such as through future waves of the COVID-19 pandemic. Therefore, the aim of this study was to estimate the per case excess direct health services costs (government perspective) for women diagnosed with incident ovarian cancer between 2006–2013, and to estimate the total national costs associated with treatment of incident and prevalent ovarian cancer in 2013.

## Methods

### Data source

This study used data from the Sax Institute's 45 and Up Study, a longitudinal study of 267,153 people from the state of New South Wales (NSW), Australia. Participants aged 45 years and over were randomly sampled from the Services Australia (formerly Department of Human Services) database, with people aged 80 years and over and rural and remote residents over-sampled. About 18% of those invited participated in the study, representing approximately 11% of the NSW population aged 45 years and over. The baseline questionnaire was completed by participants between 2006 and 2009 and written consent was provided for follow-up and linkage to routine health databases. Detailed study methods have been described elsewhere [19].

The 45 and Up Study data was linked by the Sax Institute to MBS and PBS claims records (to June 2016), supplied by Services Australia. The Centre for Health Record Linkage (http://www.cherel.org.au/) probabilistically linked the remaining datasets, including inpatient care in public and private hospitals through the Admitted Patient Data Collection (APDC; to June 2016) and emergency presentations through the Emergency Department Data Collection (EDDC; to June 2016). Linkage to the NSW Cancer Registry (NSWCR) enabled ascertainment of incident cancers up to December 2013 in NSW. Linkage to the NSW Registry of Births, Deaths and Marriages (RBDM) provided date of death data to June 2017 and linkage to the Australian Coordinating Registry's Cause of Death Unit Record File provided cause of death data to December 2015. The probabilistic matching process is known to be highly accurate (false-positive and false-negative rates <0.4%) and a detailed description of the linkage process has been reported previously [20].

The conduct of the Sax Institute's 45 and Up Study has been approved by the University of New South Wales Human Research Ethics Committee, and the NSW Population Health and Health Services Research Ethics Committee approved this specific analysis (HREC/14/CIPHS/54).

### Study population

We identified ovarian cancer cases as participants with a NSWCR notification of a first primary cancer after completion of the 45 and Up Study baseline questionnaire, with no evidence of a prior cancer (NSWCR or self-reported). The definition of "ovarian cancer" in this paper includes cancers of the ovary (International Classification of Diseases version 10 (ICD-10) code C56), fallopian tube (C57.0, C57.8), and peritoneum (C48.1, C48.2), in alignment with previous leading international studies [21]. We also restricted the sample to cases with

morphology classified as epithelial, or carcinosarcoma (malignant mixed Müllerian tumours), excluding cases with other or unspecified morphology (n = 6). Participants with borderline ovarian cancers (ICD-10 code D39.10) were not included.

Cases were matched to controls who had no NSWCR notification, no history of self-reported cancer in the baseline questionnaire and who were alive at the time the potential matched case was diagnosed with ovarian cancer. Four controls were matched to each case by sex, age (±5 years), local government area (LGA) and smoking history (3 categories: never/ex-smoker quit >15 years, ex-smoker quit ≤15 years, or current). Matching variables were ascertained from the baseline questionnaire and, where possible, missing values were ascertained from other datasets for the same person, otherwise participants were excluded from analysis. Participants with any record of healthcare coverage by the Department of Veterans' Affairs (DVA clients) were excluded as a portion of their health care costs are met by the DVA and their PBS claims data are therefore incomplete, as were cases first diagnosed after death (Fig 1). We also examined the distribution of cases and controls by remoteness of residence (Accessibility/Remoteness Index of Australia), self-reported health insurance status at baseline, and the Charlson Comorbidity Index (calculated from APDC diagnosis codes recorded at hospitalisations in the five years prior to diagnosis) [22].

## Study period

The study period for health care utilisation data was January 2006 to June 2016, with ovarian cancers diagnosed to December 2013, vital status to June 2017, and cause of death data available to December 2015.

## Variable definitions and costs determination

The detailed methods for ascertaining costs have been described previously [16]. In brief, costs were based on individuals' inpatient hospital and emergency department records (APDC and

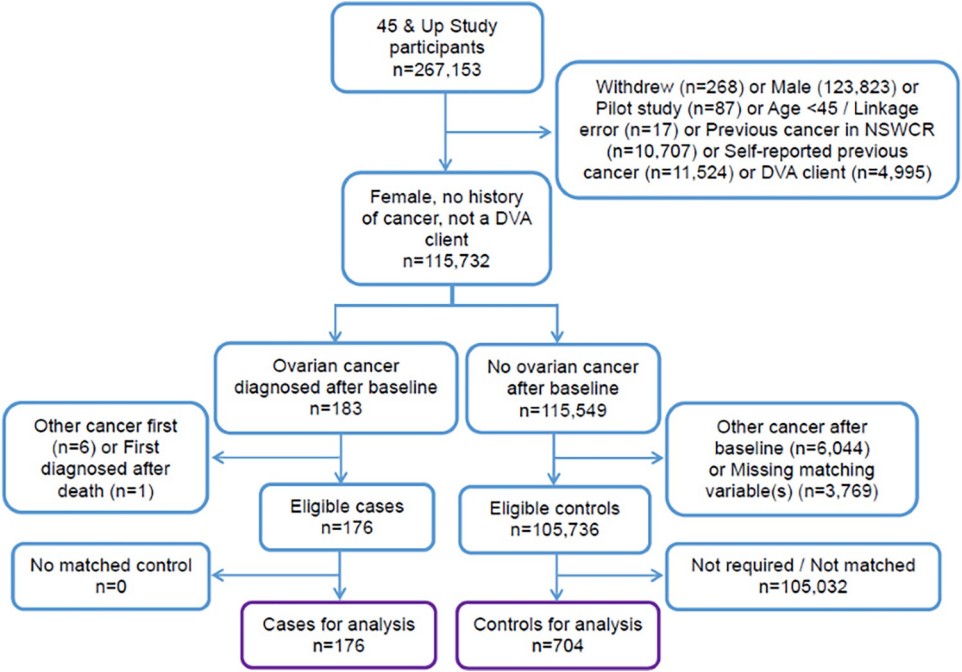

**Fig 1. Cohort selection diagram for cases of ovarian cancer ascertained in the NSW Cancer Registry to 2013 and controls in the 45 and Up Study (2006–2009).**

EDDC, respectively), subsidised prescription medicines captured in the PBS and subsidised medical services captured in the MBS. Only drugs and therapies that were subsidised through the PBS or MBS during the period of Jan 2006 –Jun 2016 were included in the total healthcare system costs. Data was not collected for other costs associated with ovarian cancer during the study period including off-label medications. The total healthcare system cost was the sum of all these components, standardised to 2013 Australian dollars using the health index from the Australian Bureau of Statistics' consumer price index, so that costs across different years were comparable.

Excess costs due to ovarian cancer were determined for each case by taking their costs and subtracting the average costs for their matched controls. Costs for each case were weighted according to disease stage at diagnosis to match the disease stage distribution for all ovarian cancers diagnosed in NSW during 2013–2017 [23]. Costs were calculated for several phases of care relative to diagnosis and death, including annual and monthly costs. Phases of care costs were defined as follows:

*Initial phase*: If the patient survived >2 years, then the first year after diagnosis was designated the initial phase. If the patient survived >1 year but < = 2 years after diagnosis, then the initial phase was the period from diagnosis until the start of the 12-month terminal phase.

*Continuing phase*: If the patient survived >2 years after diagnosis, then the period between the end of the initial phase and the start of the terminal phase, or 30 June 2016 if the case was still alive at 30 June 2017, was designated the continuing phase. The costs for this phase were calculated as an annual rate for each individual according to the length of their continuing phase. To avoid incorrect over-inflation, if the continuing phase was <3 months, then the case-control group were not included in this phase (i.e. those surviving 2–2.25 years, which was 2% of cases).

*Terminal phase*: For patients who died of any cause before July 2016, the final year up to and including the death date was designated the terminal phase. If the case survived <1 year after diagnosis then the terminal phase started at the diagnosis date and the patient had no initial phase.

Due to the differing treatment recommendations and survival for patients with localised disease compared with metastatic disease at diagnosis, excess costs for cases are also presented by stage of disease at diagnosis (distant vs. local/regional) for each phase of care. Small case numbers prohibited detailed reporting and comparisons by histological subtype, including stratification of subtype and stage at diagnosis, however results for serous epithelial carcinoma (as the most common) are reported.

We estimated the total healthcare costs for ovarian cancer incurred in Australia in 2013 for incident and prevalent cancers diagnosed in 2009–2013, using annual excess costs per case, and population-wide incidence, mortality, and survival rates for 2009–2013 as described previously [16]. In brief, we first considered the national number of ovarian cancer cases diagnosed in 2013 and applied the annual per-case costs for the year after diagnosis. Then we considered the national number of ovarian cancer cases diagnosed in 2012, used the 1-year survival to calculate the number of these alive in 2013, and applied the mean costs for the second year after diagnosis. We repeated the process up to new cases diagnosed in 2009, with 4-year survival and mean costs for the fifth year after diagnosis. The totals were combined to estimate the total healthcare costs for 2013. The same process was used to assign people to their phase of care during 2013, and the corresponding phase costs were applied.

We used RBDM date of death data to June 2017 to estimate 5-year all-cause survival, and COD-URF (cause of death unit record file) data to December 2015 to estimate 5-year cancer-specific survival (censored at December 2015). The Kaplan-Meier method was used to allow for different lengths of follow-up time.

## Results

There were 176 eligible women diagnosed with ovarian cancer in the 45 and Up Study cohort during 2006–2013, with a median age of 68 years at diagnosis. Among these women, serous carcinoma was the most common morphology type (Table 1). Seventy percent of women were classified as having distant metastases at the time of diagnosis. All cases were matched with four cancer-free controls.

**Table 1. Demographic characteristics of ovarian cancer cases in the 45 and Up Study (2006–2009) diagnosed to 2013.**

| Characteristic | Cases (n = 176) N (%) | Controls (n = 704) N (%) |
|---|---|---|
| *Age at diagnosis* | | |
| 45–59 | 46 (26) | 183 (26) |
| 60–69 | 56 (32) | 230 (33) |
| 70–79 | 41 (23) | 161 (23) |
| 80+ | 33 (19) | 130 (18) |
| *Stage* | | |
| Localised | 17 (10) | - |
| Regional | 27 (15) | - |
| Distant metastases | 123 (70) | - |
| Unknown | 9 (5) | - |
| *Histology* | | |
| Serous | 114 (65) | - |
| Mucinous | 5 (3) | - |
| Endometrioid | 6 (3) | - |
| Clear cell | 13 (7) | - |
| Adenocarcinoma NOS | 23 (13) | - |
| Other specified carcinoma[a] | 6 (3) | - |
| Unspecified carcinoma / Other specified cancer[b] | 9 (5) | - |
| *Survival[c]* | | |
| Death within 1 year | 26 (15) | <10 (1) |
| Death in >1-≤2 years | 23 (13) | <5 (<1) |
| Death in >2-≤3 years | 21 (12) | <10 (1) |
| Survived >3 years | 106 (60) | 690 (98) |
| *Place of residence* | | |
| Major cities | 87 (49) | 340 (48) |
| Inner regional | 64 (36) | 268 (38) |
| Outer regional/Remote | 25 (14) | 96 (14) |
| *Insurance status* | | |
| Private health insurance | 102 (58) | 424 (60) |
| Concession card | 48 (27) | 159 (23) |
| No insurance | 22 (13) | 106 (15) |
| Missing/Unknown | 4 (2) | 15 (2) |
| *Charlson Comorbidity Index* | | |
| 0 | 162 (92) | 644 (91) |
| 1+ | 14 (8) | 60 (9) |

[a] Includes transitional cell carcinoma, adenocarcinoma with mixed subtypes, and mixed cell adenocarcinoma.

[b] Includes Mullerian mixed tumour, and carcinosarcoma with no further specification.

[c] Survival for controls calculated from the date of diagnosis for their matched case.

**Table 2. Mean excess costs per person, by year pre-/post-diagnosis and phase of care, for people in the 45 and Up Study diagnosed with ovarian cancer in 2006–2013.**

| | > 0–1 year pre-diagnosis | | 0–1 year post-diagnosis | | >1–2 years post-diagnosis | | >2–3 years post-diagnosis | | >3–4 years post-diagnosis | | >4–5 years post-diagnosis | | Initial phase | | Continuing phase | | Terminal phase | |
|---|---|---|---|---|---|---|---|---|---|---|---|---|---|---|---|---|---|---|
| | N | Mean (SD) | n | Mean (SD) | n | Mean (SD) | n | Mean (SD) | n | Mean (SD) | n | Mean (SD) | n | Mean (SD) | n | Mean (SD) | n | Mean (SD) |
| Overall | 176 | $3,566 ($16,888) | 176 | $42,469 ($30,986) | 150 | $14,922 ($24,100) | 116 | $13,133 ($25,295) | 76 | $8,164 ($24,214) | 44 | $10,394 ($23,835) | 150 | $40,556 ($28,904) | 121 | $9,514 ($19,084) | 93 | $49,208 ($36,209) |
| Distant | 123 | $1,949 ($10,120) | 123 | $50,292 ($31,098) | 98 | $19,164 ($24,546) | 69 | $20,016 ($26,929) | 39 | $14,193 ($24,661) | 18 | $5,086 ($15,541) | 98 | $48,327 ($28,919) | 73 | $13,814 ($22,068) | 81 | $49,578 ($35,101) |
| Localised / Regional | 44 | $2,234 ($6,120) | 44 | $33,221 ($25,970) | 44 | $7,629 ($21,816) | 41 | $4,128 ($16,598) | 33 | $643 ($18,846) | 25 | $12,467 ($28,547) | 44 | $32,937 ($25,763) | 43 | $4,234 ($11,257) | 7 | $58,312 ($27,107) |
| Serous | 114 | $2,151 ($9,135) | 114 | $47,254 ($28,104) | 108 | $15,835 ($23,167) | 84 | $16,185 ($20,786) | 55 | $12,852 ($25,689) | 30 | $10,976 ($26,463) | 108 | $44,692 ($27,677) | 90 | $11,977 ($19,277) | 56 | $52,720 ($27,860) |

Includes women alive at the start of each time period. The results by disease stage do not include n = 9 women with unknown disease stage. Costs reported in 2013 Australian dollars. SD: standard deviation.

The cases were estimated to have a 5-year cancer-specific survival of 51% (55% when weighted to match the NSW stage distribution) and 5-year all-cause survival of 44% (49% when weighted). Forty percent of cases died within 3 years of diagnosis, compared with 2% of controls. Of the 15% of cases who survived less than 1 year, almost all had distant metastases at diagnosis (except for one with unknown stage). There was no difference between cases and controls in their Charlson Comorbidity Index.

During the study period, all women had at least 2.5 years of potential follow-up for health-care utilisation, with a median of 5.3 years and a maximum of 9.7 years. The mean cost from one year pre-diagnosis until two years after diagnosis was approximately $58,000 per patient, including those who died in the first year after diagnosis.

The mean excess direct health service costs per case, by year from 1-year pre-diagnosis to 5 years post diagnosis are summarised in Table 2. Mean excess costs were largest in the year following diagnosis, at $42,469 per person, with costs generally decreasing each subsequent year. Excess costs in the year prior to diagnosis were commonly for imaging procedures (e.g. CT scans, ultrasounds) and specialist consultations. By phase of care, the mean excess costs of care were $40,556 per person in the initial phase after diagnosis, dropped to $9,514 per year in the continuing phase and peaked in the terminal phase at an average of $49,208 per person. The excess costs appeared higher for cases with distant metastatic disease at diagnosis in both the initial treatment and continuing care phases (Fig 2). Analysis by histological subtype could not be done due to small numbers of rarer subtypes and effects of stage at diagnosis could not be disentangled. However, Table 2 includes costs for serous carcinoma as the more common subtype (n = 114, 77% with distant metastases at time of diagnosis).

The distribution of excess costs by health service type and phase of care is shown in Fig 3. Hospital admissions accounted for the greatest proportion of all direct health care costs (52–68%) at all phases of care. Among hospitalisations, the highest costs were for surgical procedures such as hysterectomy, and the administration of chemotherapy. The highest costs in the MBS were for imaging procedures, GP and specialist physician consultations, and pathology, while in the PBS the highest costs were for chemotherapeutic medicines such as paclitaxel, doxorubicin, and carboplatin.

Based on the number of all ovarian cancers diagnosed in Australia and the national survival rates, we estimated that 4,700 women diagnosed in Australia in 2009–2013 were alive in 2013, with 23% in the initial phase, 61% in the continuing phase, and 17% in the terminal phase.

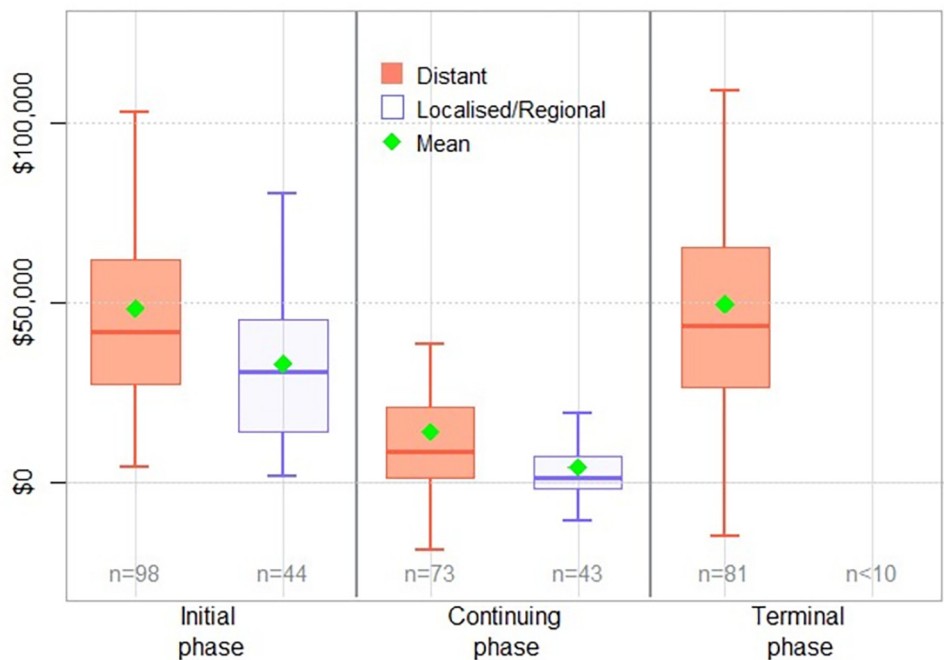

**Fig 2. Excess costs by phase of care and stage at diagnosis.** Results not shown for the terminal phase localised and regional stage cases due to the low number of deaths in this group.

Using the annual excess costs estimated in this study, the overall direct healthcare costs of ovarian cancer in these women totalled 2013AUD$99 million in 2013 alone (equivalent to 2021AUD$132 million based on the health index of the Australian consumer price index), with 40% of costs for people in the initial phase, 25% for those in the continuing phase, and 35% for those in the terminal phase.

## Discussion

We estimated that the excess health service costs for ovarian cancer in 2013 in Australia totalled approximately $99 million, giving an update to the previously reported $25 million in 2000–2001 [24]. High costs in the initial phase, and continuing phases of care compared to other cancers [16], suggest ovarian cancer is a relatively costly cancer to treat and monitor. Hospital admissions (including public and private facilities) accounted for the greatest proportion of costs in all phases of care.

Our previous cross-cancer study using the same methods and data sources as this work found that average costs of all cancers by phase were $28,719 (SD $32,151) for the initial treatment phase, $4,474 (SD $17,390) per year in the continuing care phase and $49,733 (SD $49,861) for the terminal care phase [16], compared with $40,556 (+41%), $9,514 (+113%) and $49,208 (-1%), respectively, for ovarian cancer presented here. Compared to mean estimated costs per case for the 10 most common cancers, ovarian was the 3rd most costly cancer in the initial and continuing phase [16]. Interestingly, the annual costs during the continuing care phase for ovarian cancer appeared twice as high as the estimated average of all cancers. This is partly due to the relatively high proportion of women with distant metastatic disease at diagnosis (~60%), who have higher continuing care phase costs ($13,814 compared to $4,884 for localised/regional). The continuing care phase costs might also be higher for ovarian cancer due to the relatively high rate of recurrence and progression to metastatic disease. While

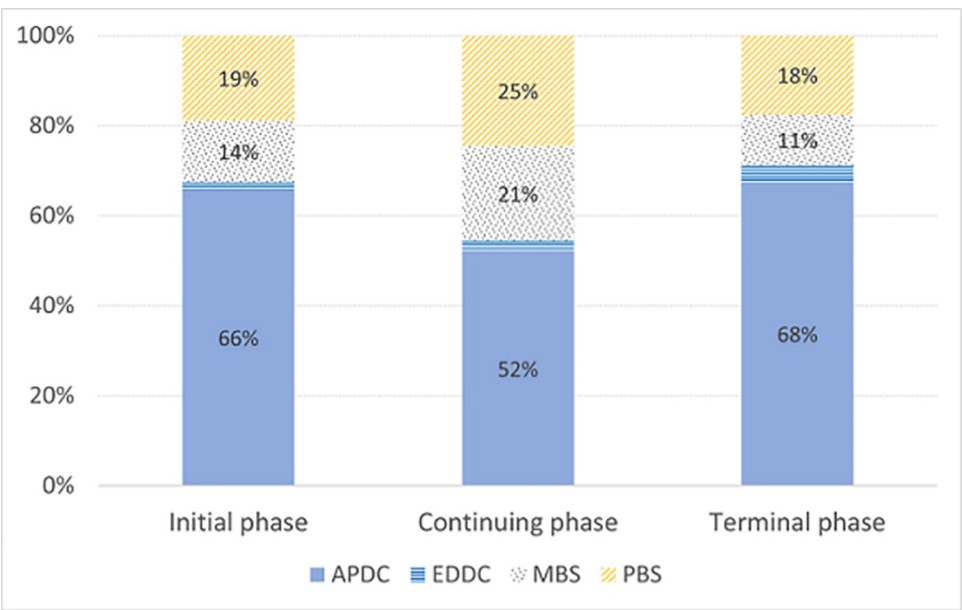

**Fig 3. Excess costs by health service type and phase of care for 45 and Up Study participants diagnosed with ovarian cancer in 2006–2009, using healthcare data to 2013.** APDC: Admitted Patient Data Collection; EDDC: Emergency Department Data Collection; MBS: Medicare Benefits Schedule; PBS: Pharmaceutical Benefits Scheme.

information on cancer recurrence in participants was not available, other studies suggest 60–80% of women with ovarian cancer relapse in 6 to 24 months following initial treatment [25], which would likely also contribute to the higher costs associated with ovarian cancer during the continuing phase. Treatment complications, adverse events and side effects also have cost implications [26], and an Australian study of women receiving treatment for ovarian cancer found that 41% of participants were hospitalised for adverse events while on chemotherapy, and 34% had surgical complications [18].

This study captured costs associated with ovarian cancer treatment for women diagnosed up to 2013, as these were the most recent data available to us at the time of analysis. However these reported costs will likely have changed since that time, through changes in clinical recommendations, licencing of new medications and technologies, and their utilisation and uptake through the health system [27, 28]. For example, changes occurring during the health care utilisation period of this study include PBS listing of bevacizumab in 2014, although there was very little take-up in this cohort. Since the study period, recommendations for genetic testing have evolved to include most women with ovarian cancer [29, 30], mainstreaming of genetic testing, and inclusion of a test to detect germline pathogenic *BRCA1* or *BRCA2* genetic variants in the MBS from 2017, at a cost of ~$1,000 per eligible person [31]. There are also new recommendations for the use of the PARPi, olaparib, based on the results of this test. Olaparib was not listed on the PBS during the study period but is now PBS listed and subsidised for women with newly diagnosed ovarian cancer who have a *BRCA* pathogenic variant [32]. This inclusion will result in additional direct health service cost of approximately $6,890 per eligible person per month [33].

Advances in personalised treatment for ovarian cancer may extend beyond PARPi in the future, with trials underway in Australia through ANZGOG and the OASIS initiative [34]. Once the molecular basis for ovarian cancer is better understood (such as through studies including the International Cancer Genome Consortium [35, 36], the Australian Ovarian

Cancer Study [37], the Ovarian Cancer Association Consortium [38] and the Ovarian Tumour Tissue Analysis consortium [39]), personalised treatment may become standard treatment for ovarian cancer, leading to potentially higher costs albeit in targeted subgroups of the patient population, or lower costs from reduced use of ineffective treatments and a reduction in side effects and complications.

We focused here only on the direct health services costs associated with an ovarian cancer diagnosis. Only women diagnosed with ovarian cancer were included and therefore any costs associated with risk-reducing surgeries [40] or genetic testing of relatives were not incorporated into this study, though they are health service costs attributable to ovarian cancer. It is also important to note that this study did not take into account any non-health system costs, including out-of-pocket health costs [41, 42], expenses and productivity costs for women with ovarian cancer, their families and carers [43]. Therefore, it is critical to further investigate the full costs and their impacts at an individual, family and societal level.

The strengths of this study include its population-based perspective, as opposed to hospital level recruitment and investigation, and the comprehensiveness of linked data and health services incorporated into the cost analysis. However, there were some differences between the cases in this cohort and all ovarian cancer cases in Australia. The median age at diagnosis in the study sample was 68 years, which is slightly older than a previous national report of 65 years [17]. This is likely due to the age of recruitment into the study (age 45 and above) and therefore exclusion of women who develop ovarian cancer at a younger age, although women aged <45 years account for 11% of incident ovarian cancers in Australia so it would have limited impact overall [44]. Younger cases are more likely to have early stage and/or non high-grade serous cancers and may have fertility-sparing treatment. Overall, those aged <40 years account for only 5% of hospitalisations [17] and thus might have lower treatment costs. The proportion of serous ovarian cases was higher in this sample than the NSW average (likely also due to the exclusion of women younger than 45 years in the cohort). The distribution of stage at diagnosis for our study population was 10% for women diagnosed with localised disease, 15% regional and 70% distant metastasis, compared with 19%, 10% and 61% respectively in all of NSW [23], again likely due to age differences with our sample containing fewer younger women with histotypes that tend to be diagnosed at early stage. The stage distribution for cases in NSW is the only reported population-wide stage distribution for ovarian cancers in Australia, and is based on ovarian and fallopian tube cancers, which is slightly different to our cohort that includes a small number of primary peritoneal cancers. There have been changes to the method of classification of ovarian cancer in recent years, which could also make the associated weighting of the study sample slightly less precise, but it should still be reasonably close to real-world practice. While the 45 and Up Study cohort achieved an 18% response rate during recruitment, participants are overall healthier than the general NSW population [19]. Thus, 45 and Up Study participants with ovarian cancer may also not be representative of all people with ovarian cancer in Australia. We have re-weighted results to match the state-wide stage distribution to improve generalisability, however caution should be used in extrapolating results to the whole-of-population level in Australia. Despite the large overall cohort, the relatively small number of incident ovarian cancers resulted in imprecise estimates for some analyses, meant that analysis by histological subtype could not be conducted in detail, and precluded further investigations of costs by geographic location. Furthermore, there were relatively few patients with localised/regional stage ovarian cancer with terminal phase information. Our prior work in colorectal and lung cancer found no clear trend in terminal phase costs by cancer stage at diagnosis [45, 46], and this could also be the case for ovarian cancer. A limitation of our dataset was that it precluded us performing detailed analysis on this aspect of ovarian cancer costs.

Through this study we have demonstrated the high costs of ovarian cancer and discussed their likely increase by 2021, and into the future. Although a proven challenge, demonstrated by the recent UK Collaborative Trial of Ovarian Cancer Screening (UKCTOCS) findings [3], investments in prevention and early detection may become more attractive as disease treatment costs increase. Comprehensive health service costs data, such as those produced through this study, are critical for evaluations of the cost-effectiveness of both new and existing interventions in ovarian cancer control.

## Conclusions

This study provides detailed cost estimates that can be used to evaluate the cost-effectiveness of existing and new interventions to optimise ovarian cancer prevention, early detection, and treatment. The results highlight the high healthcare costs of ovarian cancer and the importance of continued research investment, including the management of treatment side effects and personalisation to reduce the use of ineffective treatments, and ovarian cancer prevention strategies in women at high risk.

## Acknowledgments

This research was completed using data collected through the 45 and Up Study (www.saxinstitute.org.au). The 45 and Up Study is managed by the Sax Institute in collaboration with major partner Cancer Council NSW; and partners: the Heart Foundation; NSW Ministry of Health; NSW Department of Communities and Justice; and Australian Red Cross Lifeblood. We thank the many thousands of people participating in the 45 and Up Study, the Centre for Health Record Linkage for the record linkage and Services Australia, the NSW Ministry of Health, and Cancer Institute NSW for the use of their data.

## Author Contributions

**Conceptualization:** Amy Vassallo, Karen Canfell.

**Data curation:** David E. Goldsbury.

**Formal analysis:** David E. Goldsbury.

**Writing – original draft:** Amy Vassallo.

**Writing – review & editing:** David E. Goldsbury, Amy Vassallo, Marianne F. Weber, Julia Steinberg, Penelope M. Webb, Anna DeFazio, Karen Canfell.

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
