## [Decision Letter · Decision Letter 0]

17 Nov 2022

PONE-D-22-18547Health services costs for ovarian cancer in Australia: Estimates from the 45 and Up StudyPLOS ONE

Dear Dr. Goldsbury,

Thank you for submitting your manuscript to PLOS ONE. After careful consideration, we feel that it has merit but does not fully meet PLOS ONE’s publication criteria as it currently stands. Therefore, we invite you to submit a revised version of the manuscript that addresses the points raised during the review process.

We look forward to receiving your revised manuscript.

Kind regards,

Alvaro Galli

Academic Editor

PLOS ONE

Journal Requirements:

Reviewers' comments:

Reviewer's Responses to Questions

**Comments to the Author**

1. Is the manuscript technically sound, and do the data support the conclusions?

Reviewer #1: Yes

Reviewer #2: Yes

2. Has the statistical analysis been performed appropriately and rigorously? 

Reviewer #1: Yes

Reviewer #2: Yes

3. Have the authors made all data underlying the findings in their manuscript fully available?

Reviewer #1: Yes

Reviewer #2: No

4. Is the manuscript presented in an intelligible fashion and written in standard English?

Reviewer #1: Yes

Reviewer #2: Yes

5. Review Comments to the Author

Reviewer #1: Dear Author(s),

Your paper entitled “Health services costs for ovarian cancer in Australia: estimates from the 45 and UP Study” is very relevant since health care costs of cancer are increasing and there is a large debate about the sustainability of health care systems in all developed countries. The evidence you provided might contribute to address the health care policies in order to reduce ovarian cancer burden.

While this is a valuable contribution to the field, I have some suggestions to improve the clarity of your paper and the impact of results and interpretation.

Here below you will find my suggestions subdivided into major and minor points.

Major points:

1) According to the methods-data source section, your results are based on the Sax Institute’e 45 and Up study which includes data from 18% of individuals invited to participate in the study, they represent 11% of the NSW population aged 45+. Please add a comment in the Discussion section about the issue of the response rate of the 45 and Up Study, which influences the representativeness of the study cohort and by the consequence the transferability of the cancer cost estimations to the Australian general population;

2) A major limitation of this study is the update of the cost information since the study sample includes women diagnosed with ovarian cancer up to 2013. This implies that new treatments, particularly pharmaceutical ones, are not taken into account. However, it is not clear if all high cost drugs and target therapies are subsidized by the PBS (Pharmaceutical Benefits Scheme) or not, please add a sentence clarifying this point in the methods-data source section;

Minor points:

3) In the methods section you reference to a previous paper published in 2018 by Goldsbury et al for more details about the methods to extrapolate results from the 45 and Up study to the whole Australia in 2013, however a very short description of the methodology used would help the reader to interpret the results;

4) Stage at diagnosis is demonstrated to be a cost driver, especially in the initial phase of the disease. However results by stage in the terminal phase of care are incomplete due to the low number of deaths in localized and regional cases (see Figure 2). A comment on the expected trend by stage in the terminal phase of care compared to the continuing and initial phases should be included in the discussion;

5) It is not clear if and how you used the information about the cause of death with respect to the terminal phase of care definition: are those cases dead for causes other than cancer treated as censored cases of the continuing phase of care? are they considered eligible for the terminal phase just because they died? Please clarify;

6) Figures 2 and 3 are blurred and results are not clear, there is no contrast between different categories using black and white printing, I suggest to add textures and/or labels in the box-plot and histograms in order to make them more readable;

7) In the results section, page 12, lines 245-247, you make reference to Table 2 and indicate: “The mean cost from one year pre-diagnosis until two years after diagnosis was approximately $58,000 per patient, including those who died during the period”. However, from Table 2, the mean cost is about $61,000, summing up the mean costs by phase of care of the overall category: how do you explain this inconsistency?

8) In the results section, page 15, lines 283-287, there is the total number of prevalent cases with ovarian cancer in Australia in 2013 and the corresponding total direct health care costs estimated from data of the 45 and Up study, it would be interesting to report the distribution of these cases and costs by phase of care.

Reviewer #2: I am somewhat doubtful of the utility of this study as it reports on costs prior to the introduction of new treatments which are now considered standard care. The authors make the argument that their cost data can be used for health economic evaluations but I disagree as ‘usual care’ is not captured in their data. I am struggling to see how this paper adds something new and useful to the literature in this field.

6. PLOS authors have the option to publish the peer review history of their article (what does this mean?). If published, this will include your full peer review and any attached files.

Reviewer #1: **Yes: **Silvia Francisci

Reviewer #2: No

---

## [Author Response · Author response to Decision Letter 0]

22 Dec 2022

Please see the attached response file.

---

## [Decision Letter · Decision Letter 1]

10 Jan 2023

PONE-D-22-18547R1Health services costs for ovarian cancer in Australia: Estimates from the 45 and Up StudyPLOS ONE

Dear Dr.Godsbury,

Thank you for submitting your manuscript to PLOS ONE. After careful consideration, we feel that it has merit but does not fully meet PLOS ONE’s publication criteria as it currently stands. Therefore, we invite you to submit a revised version of the manuscript that addresses the points raised during the review process.

Reviewer 2 raised some very important issues that need to be adderessed. Please, take care of them.

We look forward to receiving your revised manuscript.

Kind regards,

Alvaro Galli

Academic Editor

PLOS ONE

Reviewers' comments:

Reviewer's Responses to Questions

**Comments to the Author**

1. If the authors have adequately addressed your comments raised in a previous round of review and you feel that this manuscript is now acceptable for publication, you may indicate that here to bypass the “Comments to the Author” section, enter your conflict of interest statement in the “Confidential to Editor” section, and submit your "Accept" recommendation.

Reviewer #1: All comments have been addressed

Reviewer #2: (No Response)

2. Is the manuscript technically sound, and do the data support the conclusions?

Reviewer #1: (No Response)

Reviewer #2: Yes

3. Has the statistical analysis been performed appropriately and rigorously? 

Reviewer #1: (No Response)

Reviewer #2: Yes

4. Have the authors made all data underlying the findings in their manuscript fully available?

Reviewer #1: (No Response)

Reviewer #2: Yes

5. Is the manuscript presented in an intelligible fashion and written in standard English?

Reviewer #1: (No Response)

Reviewer #2: Yes

6. Review Comments to the Author

Reviewer #1: (No Response)

Reviewer #2: - Please state the perspective of the study up-front.

- Can the authors comment on the impact of only including women aged 45 and older in the analyses? Do younger women incur similar costs or are there different treatments indicated for ovarian cancers diagnosed at earlier ages?

- Why was Cancer Registry data only available up to 2013? From experience, more recent data is normally available.

- Table 2: it would be useful to include 95%CI or SDs for context. Also, can the authors comment on why mean excess costs in the year preceding diagnosis were higher.

- Did the authors consider modelling the effect of socioeconomic and clinical variables on the costs?

- As many of the new therapeutic agents for cancer treatment are high cost, it would be useful to understand the impact these have had on overall costs.

- The Discussion states: Our previous cross-cancer study using the same methods and data sources as this work found 312 that average costs of all cancers by phase were $28,719 for the initial treatment phase, $4,474 313 per year in the continuing care phase and $49,733 for the terminal care phase [16], compared with $40,556 (+41%), $9,514 (+113%) and $49,208 (-1%), respectively, for ovarian cancer presented here.

It would be appreciated if the authors can present these results with a measure of dispersion.

Also in the Discussion: Interestingly, the annual costs during the continuing care phase for ovarian cancer appeared twice as high as the estimated average of all cancers. This is partly due to the relatively high proportion of women with distant metastatic disease at diagnosis (~60%), who have higher continuing care phase costs ($13,814 compared to $4,884 for localised/regional).

This difference is likely due to the selection of a 12-month cut-off for the initiation phase. Can the authors please explain why this was selected. Is this an arbitrary cut-off? Maybe this is unsuitable for ovarian cancer given the high rate of patients with metastatic spread.

Given the relatively high rates of surgical complications and adverse events for this patient group, could the authors look at how this contributed to costs? Surgical complications may be identifiable through use of Condition Onset Flag in the APDC, or a subsequent admission within 30 days that was related to the index admission.

7. PLOS authors have the option to publish the peer review history of their article (what does this mean?). If published, this will include your full peer review and any attached files.

Reviewer #1: **Yes: **Silvia Francisci

Reviewer #2: No

---

## [Decision Letter · Decision Letter 2]

24 Feb 2023

Health services costs for ovarian cancer in Australia: Estimates from the 45 and Up Study

PONE-D-22-18547R2

Dear Dr. Goldsbury,

We’re pleased to inform you that your manuscript has been judged scientifically suitable for publication and will be formally accepted for publication once it meets all outstanding technical requirements.

Kind regards,

Alvaro Galli

Academic Editor

PLOS ONE

Additional Editor Comments (optional):

Reviewers' comments:

Reviewer's Responses to Questions

**Comments to the Author**

1. If the authors have adequately addressed your comments raised in a previous round of review and you feel that this manuscript is now acceptable for publication, you may indicate that here to bypass the “Comments to the Author” section, enter your conflict of interest statement in the “Confidential to Editor” section, and submit your "Accept" recommendation.

Reviewer #1: All comments have been addressed

Reviewer #2: All comments have been addressed

2. Is the manuscript technically sound, and do the data support the conclusions?

Reviewer #1: Yes

Reviewer #2: Yes

3. Has the statistical analysis been performed appropriately and rigorously? 

Reviewer #1: Yes

Reviewer #2: Yes

4. Have the authors made all data underlying the findings in their manuscript fully available?

Reviewer #1: Yes

Reviewer #2: Yes

5. Is the manuscript presented in an intelligible fashion and written in standard English?

Reviewer #1: Yes

Reviewer #2: Yes

6. Review Comments to the Author

Reviewer #1: Your paper entitled “Health services costs for ovarian cancer in Australia: estimates from the 45 and UP Study” is relevant since health care costs of cancer are increasing and there is a large debate about the sustainability of health care systems in all developed countries. The evidence you provided might contribute to address the health care policies in order to reduce ovarian cancer burden.

While this is a valuable contribution to the field, I suggested you to review the manuscript according to a list of major and minor points, which have been properly addressed in you revised version.

Reviewer #2: Thank you for addressing the comments- nothing further from me.

7. PLOS authors have the option to publish the peer review history of their article (what does this mean?). If published, this will include your full peer review and any attached files.

Reviewer #1: **Yes: **Silvia Francisci

Reviewer #2: No

---

## [Editor Report · Acceptance letter]

10 Apr 2023

PONE-D-22-18547R2 

Health services costs for ovarian cancer in Australia: Estimates from the 45 and Up Study 

Dear Dr. Goldsbury:

I'm pleased to inform you that your manuscript has been deemed suitable for publication in PLOS ONE. Congratulations! Your manuscript is now with our production department. 

Kind regards, 

on behalf of

Dr. Alvaro Galli 

Academic Editor

PLOS ONE